# Superconducting spintronic heat engine

Clodoaldo Irineu Levartoski de Araujo [1,2], Pauli Virtanen [3] ✉, Maria Spies [1], Carmen González-Orellana [4], Samuel Kerschbaumer [4], Maxim Ilyn [4], Celia Rogero [4,5], Tero Tapio Heikkilä [3], Francesco Giazotto [1] & Elia Strambini [1] ✉

Heat engines are key devices that convert thermal energy into usable energy. Strong thermoelectricity, at the basis of electrical heat engines, is present in superconducting spin tunnel barriers at cryogenic temperatures where conventional semiconducting or metallic technologies cease to work. Here we realize a superconducting spintronic heat engine consisting of a ferromagnetic insulator/superconductor/insulator/ferromagnet tunnel junction (EuS/Al/AlO$_x$/Co). The efficiency of the engine is quantified for bath temperatures ranging from 25 mK up to 800 mK, and at different load resistances. Moreover, we show that the sign of the generated thermoelectric voltage can be inverted according to the parallel or anti-parallel orientation of the two ferromagnetic layers, EuS and Co. This realizes a thermoelectric spin valve controlling the sign and strength of the Seebeck coefficient, thereby implementing a thermoelectric memory cell. We propose a theoretical model that allows describing the experimental data and predicts the engine efficiency for different device parameters.

Thermoelectricity can be observed when an electron-hole-asymmetric conductor is driven by a temperature difference. The resulting thermovoltages and thermocurrents have been widely applied for thermometry and energy harvesting[1–3]. In the current technology, semiconductors have been extensively utilized due to the large Seebeck coefficient achievable in those materials, allowing for amplitudes of up to hundreds of μV/K[4–6] and efficient energy harvesting[7]. However, semiconductor materials are not ideal for thermoelectric-based applications in some important areas of research such as aerospace and cryogenic electronics. At extremely low temperatures of space or in dilution cryostats, the carriers in semiconductors freeze out[8] and the material becomes insulating. Approaches based on quantum dot systems have been proposed and realized[6,9] showing sizable thermopower down to 0.5 K but with limitations of scalability intrinsic to zero-dimensional systems. Moreover, semiconductor properties would be drastically changed by the doping due to background cosmic particles[10].

A promising alternative based on superconductor materials can represent a step forward in thermoelectric-based technology. Differing from semiconducting materials, metals do not suffer charge freeze-out, and semiconductor-like properties are still present in the quasi-particle spectrum characterized by the superconducting gap. Still, electron-hole asymmetry is difficult to achieve in conventional superconductors due to charge neutrality constraints, unless strong non-equilibrium conditions are present[11,12]. Recently, a superconducting spin-caloritronic scheme based on spin-selective tunnel junctions was proposed, enabling the breaking of the electron-hole symmetry while keeping charge neutrality and resulting in the generation of large thermoelectric effects[13]. This prediction was confirmed by thermocurrents measured in superconducting tunnel junctions, with spin-split superconductors obtained via external fields[14], and by exchange interactions[15] present in thin superconductor/ferromagnetic-insulator (S/FI) bilayers[16–19]. So far, a thermoelectric heat engine has never been demonstrated in this family of devices despite its key

[1]NEST, Istituto Nanoscienze-CNR and Scuola Normale Superiore, Pisa, Italy. [2]Departamento de Fisica, Laboratório de Spintrônica e Nanomagnetismo, Universidade Federal de Viçosa, Viçosa, Minas Gerais, Brazil. [3]Department of Physics and Nanoscience Center, University of Jyväskylä, Jyväskylä, Finland. [4]Centro de Física de Materiales (CFM-MPC), Centro Mixto CSIC-UPV/EHU, Donostia-San Sebastián, Spain. [5]Donostia International Physics Center (DIPC), Donostia-San Sebastián, Spain. ✉e-mail: pauli.t.virtanen@jyu.fi; elia.strambini@cnr.it

role for energy harvesting at cryogenic temperatures and resulting applications for radiation detection[20,21]

Here, we implement a superconducting spin-selective tunnel junction based on a multilayer of EuS/Al/AlO$_x$/Co. A strong thermovoltage (-10 µV) is generated at sub-Kelvin temperatures (< 1 K) with a magnitude close to its upper bound dictated by the Al superconducting gap ($\Delta \simeq 200$ µeV). The resulting Seebeck coefficient is of the order of a few hundred µV/K for different temperature and magnetic configurations. A sizable work was extracted from the junction, therefore demonstrating a superconducting spintronic heat engine. Yet, the efficiency and functionality of the engine are quantified for different magnetic configurations. Finally, the implementation of a two-state thermoelectric memory cell based on the device's magnetic hysteresis is discussed.

## Results

### Sample design and non-reciprocity

The device consists of a superconducting thin film (aluminum-Al) proximitized by a ferromagnetic insulator (europium sulfide-EuS) on one side and separated from a ferromagnet (cobalt-Co) on the other side by an insulating barrier (aluminum oxide-AlO$_x$).

Figure 1 presents a micrograph of a typical device. A schematic of the four-wire measurement used for the tunneling spectroscopy is likewise shown. In the cartoon of Fig. 1b, the side view of the sample is shown with a simplified representation of its density of states (DOS) on the bottom. The tunneling conductance of the device is strongly influenced by the exchange spin-splitting of the superconductor DOS (on the left) facing the spin-split DOS of the Co counter-electrode such that strong spin filtering is expected for proper voltage bias. Such spin filtering is at the origin of the asymmetric tunneling conductance

$G(V) = dI/dV$ measured as a function of the bias voltage and magnetic field, and presented in the color plot of Fig. 1c. The characteristic asymmetry in $G(V)$ can be seen in Fig. 1d for the green line. It is compatible with a parallel (P) alignment of the magnetizations of the EuS and Co layers, visible for most of the magnetic fields explored. Only the tunneling conductance at $B \simeq 10$ mT, obtained during the sequential switching of the two ferromagnets, is characterized by an anti-parallel (AP) alignment (Fig. 1c and d orange line). By fitting the experimental data with the spin selective tunneling model[22] (see Methods section IVB for model details and continuous lines in Fig. 1d for the fit) it is possible to extract the main parameters of the device. These are the spin polarization of the tunnel barrier $P \simeq 0.5$, the exchange interaction induced in the Al layer $h \simeq 50$ µeV, the zero-temperature Al superconducting gap $\Delta_0 \simeq 195$ µeV, and the inelastic and spin-flip scattering rates $\hbar\Gamma \simeq 32$ µeV and $\hbar\Gamma_{sf} \simeq 29$ µeV at $B = -35$ mT, respectively. Notably, a large $\Gamma$ characterizes the superconducting tunnel barrier, as typically observed in junctions with ferromagnetic counterelectrodes[23,24]. Different devices were tested showing similar results with slightly different $\Gamma$, $\Gamma_{sf}$, $h$ and $P$ (see Supplementary Fig. 2). From the tunneling spectroscopy it is also possible to extract the field evolution of the tunneling magnetoresistance TMR = $\frac{R_P - R_{AP}}{R_{AP}}$, shown in Fig. 1e, and the magnetoconductance, shown in Fig. 1f. Values obtained are compatible with previous TMR measured in similar structures[23], showing a maximum at voltage biases compatible with the superconducting gap ($eV_{MAX} \simeq \pm \Delta$).

### Thermoelectric response

To quantify the thermoelectric response of the device, the thermovoltage across the junction was measured in the presence of a thermal gradient imposed across the junction. Such temperature difference is

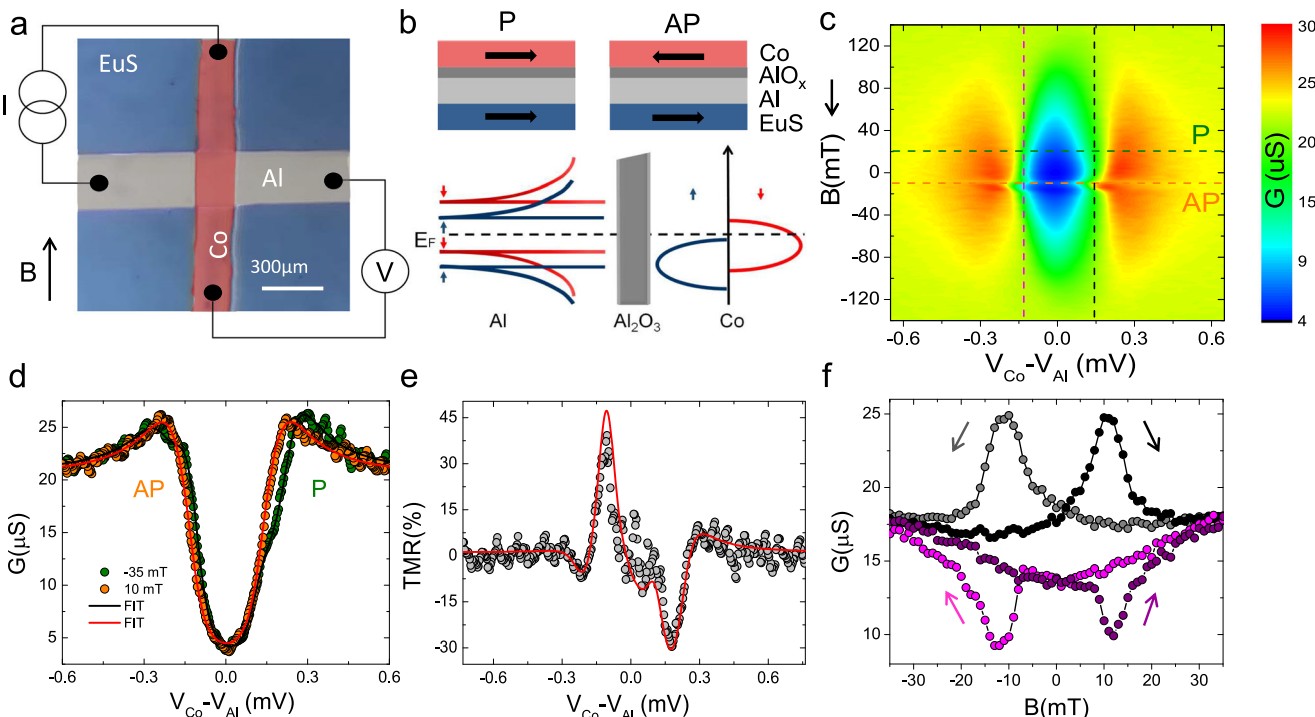

**Fig. 1 | Magneto-electric pre-characterization of the superconducting tunnel junction. a** Optical microscope image of one of the samples colored for clarity and a sketch of the 4-wire measurement set-up. The cross bar consist of EuS(13)/Al(16)/AlO$_x$(4)/Co(14), thicknesses are in nm. **b** Schematic of the sample side view (top) and simplified representation of device density of states (bottom), consisting of the spin-split Al superconducting gap (left) and spin-polarized Co 3d bands (right). **c** Contour plot of the tunneling conductance ($G(V) = dI/dV$) measured vs the external magnetic field ($B$) and the voltage drop across the junction ($V_{Co} - V_{Al}$). The sweep direction of the magnetic field is indicated by the arrow. **d** G(V) extracted from **c** showing parallel (P, green) and antiparallel (AP, orange) configurations. Continuous lines are fit to the experimental data based on the tunneling model as described in the Methods section IVB. **e** Tunneling magnetoresistance TMR=($R_{AP}$-$R_P$)/$R_P$ vs voltage drop. The red line is the theoretical expectation for the fits. **f** G(B) extracted from **c** for positive and negative voltage drops. The magnetic field sweep directions are indicated by the arrows and a hysteresis is discernible. All measurements were performed at a bath temperature $T_{bath} = 100$ mK.

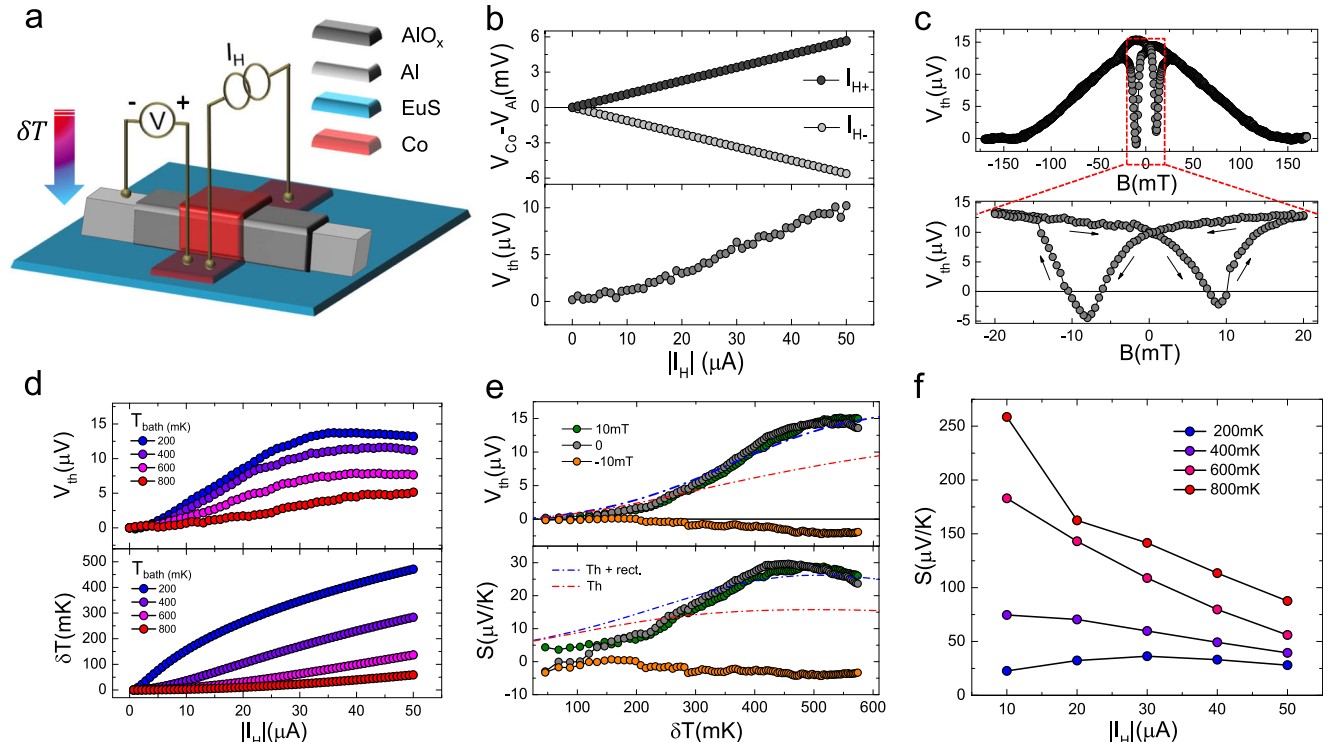

**Fig. 2 | Thermoelectric characterization. a** Scheme of the electric circuit used to quantify the thermoelectric response of the device, with the temperature difference obtained via a Joule heating current flowing through the Co strip. **b** Voltage measured vs heating current, with the positive voltmeter electrode on Co and negative on Al at B = 10 mT. The thermovoltage is obtained by subtracting the Ohmic contribution using the average $V_{th} = \frac{V(+I_H) + V(-I_H)}{2}$. **c** $V_{th}$ measured at $|I_H| = 40$ μA vs external magnetic field. **d** $V_{th}$ measured vs $I_H$ for different bath temperatures up to T = 800 mK (top panel) and temperature difference

$\delta T = T_{Co} - T_{Al}$ estimated from model fits to the tunneling spectroscopy, performed at different bath temperatures and heating currents (bottom panel).
**e** Thermovoltage (top) and Seebeck coefficient (S, bottom) vs temperature difference at selected magnetic field values measured at $T_{bath} = 100$ mK. Dash-dotted lines represent the fits to the data considering thermoelectricity with (blue) and without (red) rectification effects. **f** Seebeck coefficient S extracted at different $T_{bath}$ and $I_H$.

achieved via a Joule-heating current $I_H$ which flows through the Co strip while the Al is thermalized by the substrate at bath temperature ($T_{bath}$), according to the scheme presented in Fig. 2a. The voltage measured across the junction was then symmetrized with respect to $I_H$ ($V_{th} = \frac{V(+I_H) + V(-I_H)}{2}$) to remove the trivial ohmic contribution originating from the shared electrical paths between the voltage probe and the heating current, similarly to previous experiments on transversal rectification in superconducting tunnel diodes[25]. Differing from tunnel diodes, the larger impedance of the device makes thermoelectricity the main contribution of the voltage drop summing to rectification components. A representative example of symmetrization is presented in Fig. 2b. In the top panel the voltage measured as a function of $|I_H|$ presents the main linear evolution, while only after symmetrization (bottom panel), small deviations are visible and a clear monotonic increase of $V_{th}(I_H)$ up to $\simeq 10$ μV is observed at $I_H = 50$ μA. Above 50 μA the large power injected in the device is not fully dissipated by the substrate limiting the thermalization of the cold Al lead and resulting in a saturation or decrease of $V_{th}$. The increase of the Al temperature at large $I_H$ was confirmed by the damping of the critical current measured in the Al strip at different $I_H$ as shown in the Supplementary Fig. 3. The evolution of $V_{th}(B)$ in the external magnetic field at fixed $|I_H|$ is shown in Fig. 2c. As for the magentoconductance measurements, $V_{th}$ strongly depends on B controlling the relative orientation of the two ferromagnetic layers and by quenching superconductivity above 120 mT. In the inset, showing the central measurement range, it is possible to appreciate the sizable signal (>10 μV) present even at zero fields as a consequence of the strong ferromagnetism of the device. Moreover, a clear hysteresis is visible with a negative thermovoltage between the

coercive fields of the two ferromagnetic layers (7 mT ≲ |B| ≲ 10 mT). Such inversion of $V_{th}$ confirms the AP phase achieved between the Co and EuS ferromagnetic layers as deduced also from the hysteretic peaks observed in the magneto-conductance shown in Fig. 1f. The lower amplitude of $V_{th}$ in the AP phase concerning the P case is consistent with a weaker polarization and spin filtering of the device. This indicates a partially polarized magnetization of the EuS and Co layers in the AP phase during the non-simultaneous magnetization switching of the two ferromagnets. At higher temperatures, $V_{th}$ tends to slowly decrease as shown in the top panel of Fig. 2d with a sizable thermovoltage observed up to 800 mK. Such robustness in temperature is a consequence of the large exchange splitting of the device $h \simeq k_B \times 600$ mK extending the operation of the device to a higher temperature, see Supplementary Fig. 1. To evaluate the Seebeck coefficient from $V_{th}$ the temperature gradient across the junction $\delta T = T_{Co} - T_{Al}$ needs to be estimated. The thermal model for the device (see Method section IVB) indicates that at low heating power $T_{Al} \simeq T_{bath}$, as confirmed also by monitoring the critical current of the Al lead at different $I_H$ (see Supplementary Fig. 3). The temperature of the Co electrode is estimated from the broadening of the tunneling spectroscopy as typically done in S/I/N thermometry[26] and was measured at different $I_H$. In this case, the model needs to be extended to account also for the additional lateral voltage drop due to the presence of the heating current $I_H \neq 0$, (see Methods for details). The full model provides the estimate $\delta T(I_H, T_{bath}) = (T_{bath}^5 + b I_H^2)^{1/5} - T_{bath}$ with $b \approx 5.6 \cdot 10^{-5}$ K$^5$/μA$^2$, shown in the bottom panel of Fig. 2d. Notably, a large temperature gradient of up to 500 mK was achieved across the junction for low $T_{bath}$. Using the relation $\delta T(I_H, T_{bath})$ it is possible to

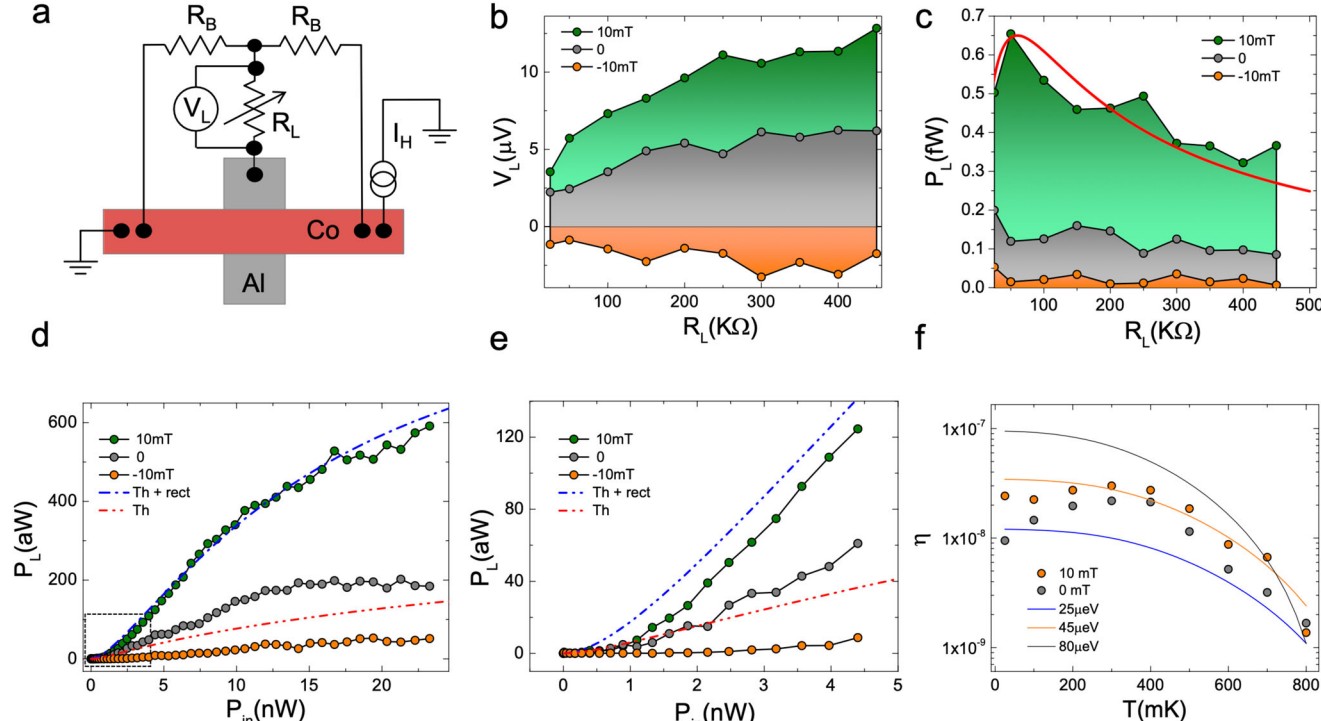

**Fig. 3 | Heat engine characterization. a** Scheme of the circuit used for the heat engine measurement. **b** Voltage developed across the load ($V_L$) measured at different load resistances ($R_L$) for $|I_H| = 40\,\mu A$ at $T_{bath} = 25\,mK$ and three magnetic field values corresponding to different regimes, i.e., P saturation at 10 mT, remanence at 0 mT, and AP configuration at −10 mT. **c** Power load ($P_L = \frac{V_L^2}{R_L}$) extracted from panel (**b**) vs $R_L$. The red curve is a qualitative fit to the 10 mT data from the maximum transferred power law $P_L = \frac{R_L V_S^2}{(R_L + R_S)^2}$, the thermoelectric voltage source

$V_S = V_{th} \simeq 12.5\,\mu V$, and an internal resistance $R_S \simeq 60\,k\Omega$. **d** Evolution of $P_L$ measured at $R_L = 150\,k\Omega$ and $T_{bath} = 25\,mK$ vs input power ($P_{in} = I_H^2 R_{Co}$). **e** Blow-up of the $P_L$ behavior of panel (**d**). Dash-dotted lines represent the fits to the data considering thermoelectricity with (blue) and without (red) rectification effects. **f** Efficiency $\eta = \frac{P_L}{P_{in}}$, measured as a function of $T_{bath}$ at $R_L = 150\,k\Omega$ and $I_H = 40\,\mu A$. Full lines are the theoretical prediction for $\eta$ for different values of $h$ (see Methods section IVB for details), and evaluated for the junction parameters used in the fit of Fig. 1d.

remap the thermovoltage in the temperature gradient $V_{th}(\delta T)$ as shown in Fig. 2e together with the resulting Seebeck coefficient $S = V_{th}/\delta T$. A Seebeck coefficient of up to 30 μV/K can be estimated both at 10 mT and at zero magnetic fields for a base temperature of 100 mK, which is on par with state-of-the-art cryogenic thermoelectric elements[9,11,12,14]. It is worthwhile to mention that by increasing the bath temperature above 500 mK, $S$ can obtain values as large as a few hundreds μV/K. Moreover, a smaller but sizable Seebeck coefficient of − 5μV/K is also visible in the AP phase obtained at $B = -10\,mT$, thus implementing a thermoelectric spin valve where the $n$-type and $p$-type Seebeck effect is controlled by the relative orientation of the device magnetic moments.

### Heat engine

Once $V_{th}$ is applied on a load resistor $R_L$, work can be extracted from the thermoelectric effect for the demonstration of a heat engine, and we can quantify the thermal-to-electrical energy conversion. The circuit used for this purpose is sketched in Fig. 3a. The junction is shunted to the ground with $R_L$ and two additional balancing resistors ($R_B = 10\,k\Omega$) have been included in the circuit in a symmetric configuration to prevent spurious leaks from the heating-current source to the load. The voltage drop ($V_L$) across $R_L$ is probed for different measurement configurations and the resulting dissipated power ($P_L = \frac{V_L^2}{R_L}$) is used to estimate the power generated by the engine, then neglecting residual thermoelectric power dissipated in the balancing resistors $R_B$. The evolution of $V_L$ measured as a function of $R_L$, under constant heating with $|I_H| = 40\,\mu A$ is presented in Fig. 3b. Three different magnetic configurations are compared, P saturation regime (green, $B = 10\,mT$), P remanence (gray, $B = 0\,mT$) and AP (orange, $B = -10\,mT$)

characterized by a negative $V_L$ as for the thermovoltage shown in Fig. 1c. In all presented field regimes $V_L$ tends to increase with an increasing $R_L$, showing a saturation towards the thermovoltage $V_{th}$ when the load resistance is above 500 kΩ, i.e., much larger than the tunnel resistance ($R_L \gg R_T \simeq 50\,k\Omega$). This is the expected behavior of an ideal voltage source $V_L = V_{th} \frac{R_L}{R_L + R_T}$, with $R_T$ as the source of internal resistance. The resulting dissipated power shown in Fig. 3c has a different non-monotonic behavior with a maximum observed for $R_L \sim R_T$. This behavior is consistent with the maximum power transfer theorem (known as Jacobi's law) predicting $P_L = I^2 R_L = \frac{V_L^2 R_L}{(R_L + R_T)^2}$ as reported in the red fit line in Fig. 3c.

The heat engine efficiency can be quantified from the ratio between the power extracted by the engine ($P_L$) and the Joule power injected to generate the thermal gradient $P_{in} = R_{Co} \times I_H^2$, where $R_{Co} = 11\,\Omega$ is the resistance of the cobalt strip at the junction. Fig. 3d,e shows $P_L$ vs $P_{in}$ for the three magnetic field configurations. At low power, $P_L$ is characterized by an almost linear increase corresponding to an efficiency $\eta = P_L/P_{in} \simeq 5 \times 10^{-8}$ that tends to decrease at high power. The temperature dependence of $\eta$ displayed in Fig. 3f shows an almost constant efficiency below 400 mK, where $T_{Co}$ saturates to the value given by the Joule heating, and a quick damping at higher temperatures. The quick damping is consistent with the expected behavior of $\eta(T)$ obtained from the theoretical model shown in Fig. 3f based on the device parameters, as described in detail in the methods section. The model shows that the main limiting factor for $\eta$ at high temperatures is the electron-phonon coupling: most of the thermal energy from the electrons is transferred to the lattice phonons, instead of being converted into thermoelectric power. The thermal conductance between electrons and lattice phonons scales as $\propto T^4$ [26,27], leading to a

 

decreasing efficiency at increasing temperatures. Yet, as observed from the theoretical model, the heat engine efficiency could be strongly enhanced by working below 8 mK for the Co electrode temperature. The threshold temperature $T_{min}$ below which electron-phonon coupling can be neglected depends on intrinsic sample parameters including the electron-phonon coupling strength in cobalt $\Sigma_{Co}$, the thickness of the cobalt layer $t_{Co}$ and the square resistance of the tunnel junction $R_\square$, $T_{min} \propto (\Sigma_{Co} t_{Co} R_\square)^{-\frac{1}{3}}$. With our sample parameters we estimate $T_{min} \simeq 8$ mK and $\eta(T_{Co}) \propto (\frac{T_{min}}{T_{Co}})^3$ (see the methods section for more details on the estimation). Additional elements that may affect the efficiency of the engine can be recognized in the non-ideal spin filtering ($P < 1$) and the nonzero Dynes parameter ($\Gamma > 0$). The former decreases the efficiency by a factor $\frac{P^2}{1-P^2} \simeq \frac{1}{3}$, the latter by one order of magnitude as shown in Supplementary Fig. 1. In other words, the efficiency of the device could be greatly increased by using more transparent junctions, and decreasing the role of electron-phonon coupling, for example, by heating Al instead of Co. The Joule heating technique used in the present work is not compatible with heating Al since the current gets converted to supercurrent inside Al. Such a problem is not present when using high-frequency radiation to heat the system.

### Heat-engine memory

It is worthwhile noting that by combining the hysteretic behavior of the thermoelectric effect as shown in Fig. 1f with the heat engine, it is possible to envision a thermoelectric memory cell. Our device structure is an original concept for a classical memory cell like a conventional magnetic random access memory stack[28]. The latter is composed of two ferromagnetic layers with different coercivity and separated by a tunnel junction. The first ferromagnetic electrode provides the electronic spin polarization in two Mott channels[29], while the second ferromagnetic layer filters the spin-polarized currents after coherent tunneling, thereby allowing high conductance in P configuration and lower conductance in the AP state. By contrast, in our memory cell, the logic states are codified by a thermoelectric voltage self-generated in the memory itself, thus not requiring a local input current for the state read-out. This advantage can strongly simplify the wiring of the memory with net benefits for scalability. In addition, it allows for local direct transduction of the electrical signal into another physical observable, for instance, a photon in the sketch of Fig. 4a. This may inspire novel methods for read-out and packaging in dense arrays.

In Fig. 4b, we present an example of the heat-engine hysteretic cycle measured in the proposed memory cell. The device is first polarized in the P state at −20 mT, and then $B$ is cycled between −20 mT to 10 mT. A clear hysteretic loop is visible with a high contrast of 10 μV between the P and AP configuration also at zero field, an important condition to operate the memory in the absence of external magnetic fields.

## Discussion

In summary, we have fabricated and characterized a superconductor-ferromagnet tunnel junction structure based on aluminum proximitized by europium sulfide and separated from a cobalt electrode by an aluminum oxide tunnel barrier. Our device shows a remarkable non-reciprocal charge transport due to electron-hole symmetry breaking induced by the spin selectivity of the junction. As a consequence, a sizable thermoelectric voltage is observed in the presence of a thermal gradient, which is achieved via a Joule-heating current flowing through the cobalt strip. The different coercivity of the two ferromagnetic layers exploited in the junction joined to the large ferromagnetic remanence of Co magnetization warrant two important features: (i) thermoelectricity is observed even at zero magnetic fields and (ii) a clear inversion of the thermoelectric effect is achieved when the EuS and Co magnetizations are antiparallel. The latter implements a spin valve for thermoelectric applications[30,31], by reversing the Seebeck coefficient from $p$-type to $n$-type. We quantified the power generated by the structure over a series of external load resistors thereby demonstrating the implementation of a superconducting spintronic heat engine. From the thermal model of the device, we identified the heat losses through the electron-phonon coupling as the main factor limiting the engine efficiency. In light of future technological applications, several strategies could be followed to increase the engine efficiency, such as decreasing the junction resistance and improving heat isolation via device suspension or by lowering the operating temperature in order to limit the electron-phonon coupling. We envision the application of such cryogenic thermoelectric elements in the implementation of sensitive self-biased detectors of electromagnetic radiation with simplified approaches for multiplexing[20,21,32]. In addition, by increasing its operation temperature above a few Kelvins the heat engine may find relevant applications also for energy harvesting in the deep space where the very low temperatures make conventional approaches somewhat ineffective. This can be achieved by replacing the Al with other superconductors with higher critical temperatures

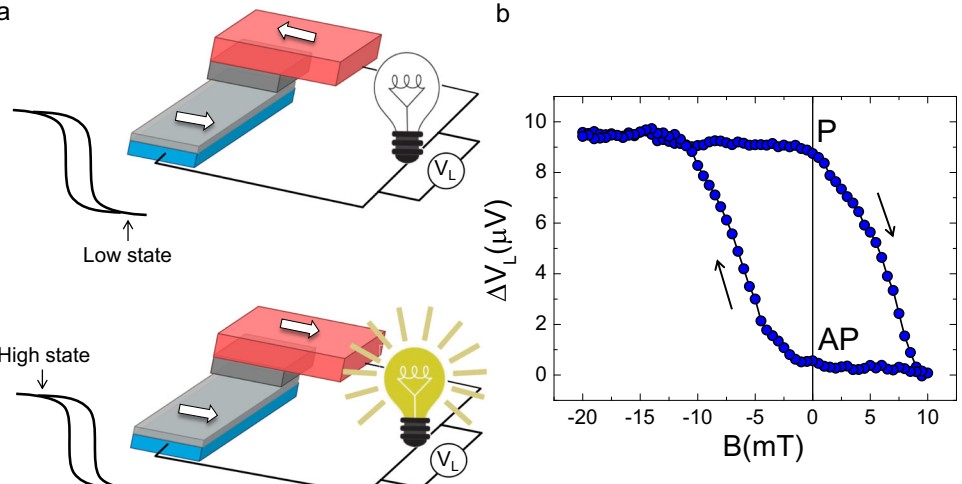

**Fig. 4 | Thermoelectric memory cell. a** Simplified scheme representing the hysteretic behavior of the thermoelectric memory cell. The low state is represented by the AP configuration between the spin-split superconductor and the ferromagnetic cobalt layer, while the high state is in the P configuration characterized by a larger voltage generated across the load resistor. **b** Evolution of the differential voltage drop across the load ($\Delta V_L = V_L^P - V_L^{AP}$) measured in the memory cell vs magnetic field $B$ with a load resistor $R_L = 150$ kΩ at $T_{bath} = 300$ mK. Note the sizable signal contrast of 10 μV that the memory cell can provide even at zero magnetic fields.

like V, Nb or NbN already integrated with EuS or GaN thin films[33–35]. In this case, engine efficiencies ranging from $10^{-4}\eta_{\text{Carnot}}$ up to $10^{-2}\eta_{\text{Carnot}}$ can be reached with particular attention in the design to limit the el-ph thermal losses and avoid inefficiencies (see Methods section IV C for details). Finally, we have shown the operation of the device as a thermoelectric superconducting memory cell. For that purpose, two main advantages are envisioned: (i) junction durability, if operated in an open circuit configuration with no detrimental currents flowing through the junction; (ii) high scalability, due to a read-out signal self-generated by the thermoelectric power.

## Methods

### Sample preparation and measurement

For the sample preparation, we first deposited 12.5 nm EuS thin film by molecular beam epitaxy on top of Si/SiO$_x$ substrates cooled at 150 K. The pressure during growth was kept in the range of $10^{-9}$ mbar to avoid any EuS oxidation and achieve a near stoichiometric EuS compound. Without breaking the ultra-high vacuum conditions, a ~ 250- µm-width Al lead was grown on top by using a metallic shadow mask. The total thickness of the Al layer was 20 nm. To form the insulating AlO$_x$ barrier the sample was exposed $3 \times 10^{-3}$ mbar of low-energy oxygen plasma created by an inductevely coupled plasma source for 5 h, resulting in a ~ 4- nm-thick AlO$_x$ layer and lowering the metallic Al thickness down to 16 nm. The subsequent cross bar geometry was realized with another shadow mask evaporation to grow a Co-lead of 14 nm thickness and ~ 200 µm width. Al and Co layers were grown with e-beam metal evaporators. Finally, before extracting the sample from the chamber, a 7 nm calcium fluoride (CaF) layer was deposited covering the whole system to avoid environmental oxidation. Samples were wire bonded with aluminum wires and mounted in a dilution fridge, where the magneto-electrical measurements were performed through low-pass filters. All the signals were amplified via low-noise voltage and current preamplifiers. For the application of the heating currents and critical current measurements, the filters were bypassed to decrease the power load that would provide cryostat excessive heating.

### Theoretical model

The current density through the Al/Co tunnel junction can be expressed as[13,22],

$$\mathcal{I} = \sum_{\sigma=\uparrow,\downarrow} G_\sigma^\square \int_{-\infty}^{\infty} dE\, N_\sigma(E)[f(E,T_S) - f(E+V,T_N)], \quad (1)$$

and depends on the voltage $V$ over the junction and temperatures $T_N$, $T_S$ on the normal (Co) and superconductor (Al) sides. Here $f(E,T)$ is a Fermi function. The current is proportional to spin-dependent conductances per square area $G_{\uparrow/\downarrow}^\square = \frac{1\pm P}{2} G_\square$, which are due to Co spin polarization and interface properties. Here, $-1 \leq P \leq 1$ is the spin polarization and $G_\square$ the junction conductance per square. The result also depends on the superconductor density of states $N_\sigma(E)$. The superconducting gap $\Delta$ in it is spin-split by an exchange field $h$ induced from EuS, but this is counteracted by spin-flip scattering with rate $\Gamma_{\text{sf}}$ and inelastic scattering with rate $\Gamma$, which we account for using methods in ref.[22]. We extract the values of $\Delta$, $\Gamma_{\text{sf}}$, $\Gamma$, $G$, $P$, and $h$ by fitting Eq. (1) to experimental $I(V)$ characteristics at $T_S = T_N$. The gap $\Delta(T,h)$ is determined self-consistently following the discussion in ref.[22], with $\Delta_0 \equiv \Delta(0,0)$ as a fit parameter. These fits and the resulting $\Delta(T)$ are illustrated in the Supplementary Fig. 4.

As the Al/Co tunnel junction resistance is high compared to the total Co wire resistance, the voltage profile along the Co wire is linear to a good approximation, $V(x) = V_0 + xI_H R_x/L_x$, where $R_x = \rho_{\text{Co}} L_x/(W t_{\text{Co}})$ is the lateral resistance of the part of the Co film (cross-section $t_{\text{Co}} \times W$, length $L_x$, resistivity $\rho_{\text{Co}}$) on top of the tunnel junction. Due to super-flow, the voltage in the Al film is spatially constant. The total tunneling current through the Al/Co junction then is

$$I_T = W \int_{-L_x/2}^{L_x/2} dx\, \mathcal{I}(V(x),T_{\text{Co}},T_{\text{Al}}) \quad (2)$$

where the Al/Co overlap has size $W \times L_x$ and $\mathcal{I}$ is the local S/I/FM junction current density–voltage relation from Eq. (1), which includes the S/FI thermoelectric effects[13]. Under these conditions, the Joule heating via current $I_H$ in Co at low temperatures is mainly limited by the electron-phonon coupling. The corresponding heat balance equation is[26],

$$L_x W t_{\text{Co}} \Sigma (T_{\text{Co}}^5 - T_{\text{bath}}^5) = R_x I_H^2, \quad (3)$$

where $\Sigma$ the electron-phonon coupling parameter of Co. We have also modeled the Al side with a similar equation, with tunneling current input power on the right-hand side and taking superconductivity into account[13,20,26]. We find that due to the high tunnel resistance, electronic heat transport across the tunnel junction is suppressed, and we can neglect the heating of the Al side. Consequently $\delta T = T_{\text{Co}} - T_{\text{Al}} = (T_{\text{bath}}^5 + bI_H^2)^{1/5} - T_{\text{bath}}$, with $b = R_x/(W t_{\text{Co}} L_x \Sigma)$. To determine the effective values of $\Sigma$ and $R_x$, a two-parameter fit of Eqs. (2),(3) is done on the experimental $dI/dV$ curves of different $T_{\text{bath}}$ and $I_H$. In addition, fits at $I_H = 0$ are used to determine the tunnel junction parameters in Eq. (1).

The circuit model of Fig. 3a together with Eqs. (1),(2),(3) form the model of the heat engine, from which efficiencies and relative contributions of the rectification and thermoelectricity can be estimated. For small $I_H$, we can expand $I_T \approx GV_{\text{th}} + P\alpha\delta T/T + \frac{G'}{24}(R_x I_H)^2$ where $G$ is the tunnel junction conductance at zero bias, $\alpha$ the thermoelectric coefficient[13], and $G'$ the zero-bias voltage derivative of the conductance, characterizing the rectification. In the tunneling model, $G' \approx ce^2 P\alpha/(2k_B^2 T^2)$ where $c \approx 1$ is a weakly temperature-dependent numerical factor. From Eq. (3) one can then deduce that for high-resistance tunnel junctions, the thermoelectric contribution dominates when $T_{\text{bath}} \lesssim [9k_B^2/(e^2\rho_{\text{Co}}\Sigma L_x^2)]^{1/3} \approx 300$ mK and $I_H \lesssim 25[k_B^5/(e^5 R_x^4 W t_{\text{Co}} \Sigma L_x)]^{1/3} \approx 20$ µA. The opposite limit of low-resistance junctions was discussed in ref.[25].

The thermoelectric coefficient $\alpha$ is related to the Seebeck coefficient by $S = V_{\text{th}}/\delta T = P\alpha/(GT)$, and its temperature and exchange field dependence was discussed in ref.[13]. The large value of $\Gamma$ in the experiment modifies the temperature and exchange field dependence, as illustrated in the theoretical prediction in the Supplementary Fig. 1.

### Efficiency estimation

The heat engine efficiency ($\eta$) can be predicted from the above model, by calculating $V_{\text{th}}$ in the presence of the load, and the corresponding $P_L$ dissipated in the load. For small temperature differences (linear response), it can be estimated following ref.[22]. The maximum efficiency with an optimally chosen load resistor is $\eta = \eta_{\text{Carnot}}(\sqrt{1+ZT} - 1)/(\sqrt{1+ZT} + 1)$, where $\eta_{\text{Carnot}} = \Delta T/T_{\text{hot}}$ is the Carnot efficiency and $ZT$ is the thermoelectric figure of merit of the device. For small $ZT$, we have $\eta \approx \eta_{\text{Carnot}} ZT/4$. For devices such as here, $ZT = \frac{S^2 GT}{\tilde{G}_{\text{th}}}$, where $\tilde{G}_{\text{th}}$ is the total heat conductance across the thermoelectric contact at zero current. In the absence of electron-phonon coupling, for small $\Gamma$ and for $k_B T \ll h$, one can show[13] that $ZT \approx P^2/(1-P^2)$. With $P = 0.5$ and $\Delta T = 300$ mK $\approx T_{\text{hot}}/2$, the corresponding ideal efficiency would be $\eta \approx 10^{-1}$. However, here $\tilde{G}_{\text{th}}$ is dominated by electron-phonon coupling, and the true $ZT$ is reduced approximatively by the ratio of thermal conductance for the electron-phonon coupling

in Co and that across the tunnel junction, i.e.,

$$\frac{G_{e-ph}^{th}}{G_T^{th}} \simeq \frac{5At_{Co}\Sigma_{Co}T_{Co}^4}{\pi^2 k_B^2 A G_\square T_{Co}/(3e^2)} = \left(\frac{T_{Co}}{T_{min}}\right)^3 \qquad (4)$$

in the normal state. Here $\Sigma_{Co} \sim 2 \times 10^8\,\mathrm{W/(m^3K^5)}$ is the e-ph heat conductivity estimated from $dI/dV$ measurements, $G_\square = 20\,\mu\mathrm{S/}$ $(250 \times 300\,\mu\mathrm{m}^2)$, Co layer thickness $t_{Co} = 14\,\mathrm{nm}$ and $T_{min} = \left(\frac{\pi^2 k_B^2 G_\square}{15 t_{Co}\Sigma_{Co}e^2}\right)^{1/3} \simeq 8\,\mathrm{mK}$. We compare the electron-phonon heat conductance to the normal-state Wiedemann-Franz heat conductance of the tunnel barrier, because of the large subgap conductance of the samples (see, for example, Fig. 3d). Suppression of $G_T^{th}$ by superconductivity of Al, and reduction of $ZT$ by the nonideal $h$ and nonzero $\Gamma$ and $\Gamma_{sf}$ produce further reductions. These factors together result to $\eta \sim 10^{-8}$ values. The decrease of $\eta$ as the temperature increases is mainly associated with the increase of electron-phonon heat conductance.

To extend the operation temperature above 3 K high-temperature superconductors like V, Nb or Nb may be used. Typically, high-temperature superconductors have larger spin orbit interactions additionally degrading the efficiency of the engine that can be compensated with special attention in the engine design. An optimal heat engine operating at 3 K can be evaluated. Thin Nb or V films replace the Al of our device. By first neglecting the spin-orbit relaxation and assuming $\Gamma \ll \exp(-\Delta/k_B T)$, $k_B T \ll h \ll \Delta$ the figure of merit $ZT$ can be estimated from Eq. (151) of ref. 20. With the same parameters of our experiment ($t_{Co} = 14\,\mathrm{nm}$, $P = 0.5$, $\Sigma_{Co} \sim 2 \times 10^8\,\mathrm{W/(m^3K^5)}$) and a larger tunneling conductance $G_\square \sim 2 \times 10^5\,\mathrm{S/(m^2)}$, $h = 0.3\Delta$ and $\Delta_{Nb} = 1.3\,\mathrm{meV}$ we estimate $ZT \approx 0.004$ and hence $\eta \approx 0.001\eta_{Carnot}$. Introducing a spin-orbit scattering time $\tau_{so} = 0.2\,\mathrm{ps}$ typical for Nb[36], there is a further factor 10 reduction due to spin-orbit scattering as can be estimated from Fig. 30 of ref. 20 with $\hbar/(\tau_{so}k_B T_{c0}) \approx 4$.

On the other hand, heating of the superconductor with local current injection is in principle possible in different device structures. In that case, the role of electron-phonon scattering is strongly reduced, and the efficiency is limited by the subgap density of states, only. In this case is possible to reach $ZT \approx 0.1P^2/(1-P^2) \approx 0.04$ and efficiency $\eta \sim 0.01\eta_{Carnot}$, see Fig. 30 of ref. 20.

### Reporting summary

Further information on research design is available in the Nature Portfolio Reporting Summary linked to this article.

## Data availability

The data that support the findings of this study are provided in the Supplementary Data 1 file. More information is available from the corresponding author E.S. upon reasonable request.

## Code availability

The codes that support the findings of this study are available from the corresponding author P.V. upon reasonable request. The code used for IV and self-consistent calculations is available at https://doi.org/10.17011/jyx/dataset/93202.

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

## Acknowledgements

C.I.L.A., P.V., T.T.H., and F.G. acknowledge funding from the EU's Horizon 2020 Research and Innovation Program under Grant Agreement No. 800923 (SuperTED). M.S. and E.S. acknowledge funding from the European Union's Horizon 2020 research and innovation program under the Marie Skłodowska Curie Action IF Grant No. 101022473 (Super-CONtacts). F.G. and E.S. acknowledge the EU's Horizon 2020 Research and Innovation Framework Program under Grant Agreement No. 964398 (SUPERGATE), No. 101057977 (SPECTRUM), and the PNRR MUR project PE0000023-NQSTI for partial financial support. C.I.L.A. acknowledges Brazilian agencies FINEP, FAPEMIG APQ-04548-22, CNPq, and CAPES (Finance Code 001). C.G.O., S.K., M.I. and C.R. acknowledge financial support by the Spanish MCIU/AEI/10.13039/501100011033, and by the European Union "NextGenerationEU"/PRTR (grants No. PID2022-138750NB-C22 and TED2021-130292B-C42). T.T.H. and P.V. acknowledge the funding from the Research Council of Finland (grant no. 354735).

## Author contributions

C.I.L.A., M.S., and E.S. experimented and analyzed the data. P.V and T.T.H. provided theoretical support. C.G.O., S.K., M.I., and C.R. fabricated the samples. E.S. conceived the experiment together with F.G. T.T.H., C.I.L.A., M.S., P.V., and E.S. wrote the manuscript with feedback from all authors.

## Competing interests

The authors declare no competing interests.
