## [Peer Review File · Nature Communications]

REVIEWER COMMENTS

Reviewer #1 (Remarks to the Author):

The article is dealing with superconducting spintronic heat engine. Heat engines are key devices that convert thermal energy into usable energy. It is important to realize a superconducting spintronic heat engine for superconducting spintronics. In this manuscript, the thermovoltage, the power generated by the engine and the heat engine efficiency have been measured. The topic is impressive and interesting, the manuscript can be published with some revision.

1, The thermoelectric effect in superconducting junctions has been extensively studied. The thermoelectric effect can help us to determine some fundamental properties of superconductivity such as pairing properties and Andreev bound states. In this paper the authors study the work done by superconducting heat engine and the efficiency of the engine for bath temperatures ranging from 25 mK up to 800 mK. I'm wondering what specific applications there are for studying the work done by a heat engine and its efficiency at these extreme low temperatures. Because this journal has a broader readers, the authors should explain the significance of this work in detail.

2, The captions of figures 1 and 3 are too long and the author could have put some details into the text.

3, Equation (1) is a bit too simple and too many parameters Δ , Γ , G , P and h are fitted through it, which will affect the accuracy of the results. The density of states of superconductors is theoretically calculated by self-consistency and varies with temperature, none of which is given in the paper. The authors can give the details in the supplementary material.

4, Extended Data Figures 2 and 3 are given but are not illustrated or discussed in the paper or after Extended Data Figures 2 and 3.

In general, the authors have studied superconducting spintronic heat engine, which is very important and timely, but they need to give a more detailed and rational theoretical explanation of the results obtained.

Reviewer #2 (Remarks to the Author):

The current manuscript has studied on the superconducting spintronic heat engine consisting of a ferromagnetic insulator/superconductor/insulator/ferromagnet tunnel junction (EuS/Al/AlOx/Co), which shows that the sign of thermoelectric voltage can be reversed depending on the parallel or anti-parallel states between ferromagnetic EuS and Co. Furthermore, the efficiency of the engine is evaluated for different magnetic configurations and thermoelectric memory cell has been discussed based on the device magnetic hysteresis. The theoretical model has been proposed to predict the engine efficiency with different parameters. Before providing my general opinion on the work, I wish to raise a few concerns:

1)The author claimed that a strong thermovoltage can be obtained at sub-Kelvin temperatures and the resulting Seebeck coefficient can reach up to the order of few hundred of $\mu\text{V}/\text{K}$ at different temperature and magnetic configurations. This should be the most important part for this work. However, such magnetic-order-dependent thermoelectric behavior have been discussed in the previous publications, such as Phys. Rev. Lett. 130, 237001 (2023) and Phys. Rev. Lett. 116, 097001 (2016).

2)The authors have evaluated the engine efficiency for the so called the superconducting spintronic heat engine system, which is about 10^{-8} . The small energy conversion efficiency, in contrast to the conventional heat engine, would limit the further applications for the energy harvesting. Also, the discussion should be added to improve the working temperature by substituting superconductor Al with other high-temperature superconducting materials.

Other minor comments:

- The labels in the extended data Fig. 2 should be distinguished.
- The first letter of journal name in the references should be capital.
- The units and the number should be separated.

Overall, the results are important but some of them are not completely new in the paper, although they are valid. Therefore, I cannot recommend for the publication in Nature Communications.

Reviewer #3 (Remarks to the Author):

The manuscript proposes a fabrication and also theoretical investigation of the superconducting spintronic heat engine consisting of a ferromagnetic insulator/superconductor/insulator/ferromagnet tunnel junction (EuS/Al/AlOx/Co). Also, the implementation of the thermoelectric memory cell is reported due to the reverse of the sign of the generated thermoelectric voltage according to the parallel or anti-parallel orientation of the two ferromagnetic layers.

The work is interesting specially for thermoelectric-based applications in cryogenic electronics where at the extremely low temperatures, the semiconductor materials are not anymore suitable for energy harvesting.

Although the work sounds, but it lacks perfection considering the following concerns:

- The lack of commas “,” is felt throughout the whole manuscript, leading to the misdirect concepts.

For example:

P4, line 61: A sizable work was extracted by the junction therefore demonstrating a superconducting spintronic heat engine.

This sentence without any comma is misleading and hard to follow

- - Very long sentences:

For example:

P4, lines 82-87: By fitting the experimental data with the spin selective tunneling model [22] (see Methods section VII B for model details and continuous lines in Fig. 1d for the fit) it is possible to extract the spin polarization of the tunnel barrier $P \approx 0.5$, the exchange interaction induced in the Al layer $h \approx 50 \mu\text{eV}$, the Al superconducting gap $\Delta \approx 195 \mu\text{eV}$, and the inelastic and spin-flip scattering rates $\hbar\Gamma \approx 32 \mu\text{eV}$ and $\hbar\Gamma_{\text{sf}} \approx 29 \mu\text{eV}$, for $B = -35 \text{ mT}$,

respectively.

- Verification/validation Cases:

For example:

P.5, lines 93-95: Values obtained are compatible with previous TMR measured in similar structures [23], showing a maximum at voltage biases compatible with the superconducting gap ($eV_{MAX} \approx \pm\Delta$).

It is important that the author name the specific case of the verification clearly and mention the approach. It is better to show the obtained result and the case of the verification/validation in the same plot, not just citing the work. In the present form one should go and check the reference [23] and find out that only the behavior is the same while the authors claim the same structure is investigated.

[23] B. Li, G.-X. Miao, and J. S. Moodera, Observation of tunnel magnetoresistance in a superconducting junction with zeeman-split energy bands, Physical Review B 88, 161105 (2013).

Or the case of validation also from the present manuscript:

P 5,6, lines 119-121: Hysteresis in the magnetic field is visible, with a maximum signal at $|B| \approx 20$ mT vanishing above 120 mT due to the quenching of superconductivity, as observed also in the tunneling spectroscopy measurements reported in fig. 1c.

In fact, the manuscript lacks the strong validation/verification part. To make the manuscript clearer and more reliable, I suggest the author write a section entitled "verification/validation" to mention the cases. The results and the validation/verification cases are mixed and this makes the manuscript difficult to be followed.

- Some sentences are needed to be more explained

The manuscript presents lots of information but somewhere it lacks the clear justifications.

For instance,

1. The behavior of the efficiency appeared in Fig. 3f is not well justified.
 2. I do not see a clear explanation about the claim that "heat losses through the electron-phonon coupling as the main factor limiting the engine efficiency".
 3. More explanation of the claim "by scaling their heat engine to the large areas, they may find relevant applications for energy harvesting in the deep space". How does this work?
- In brief, I recommend the publication of the manuscript if the mentioned comments are fulfilled.

We thank the referees for a careful reading of our manuscript and the comments aimed at improving it. Our detailed response and actions concerning the reviewer reports is below. Detailed formatting of the reviewer reports is ours.

Referee 1

The article is dealing with superconducting spintronic heat engine. Heat engines are key devices that convert thermal energy into usable energy. It is important to realize a superconducting spintronic heat engine for superconducting spintronics. In this manuscript, the thermovoltage, the power generated by the engine and the heat engine efficiency have been measured. The topic is impressive and interesting, the manuscript can be published with some revision.

1, The thermoelectric effect in superconducting junctions has been extensively studied. The thermoelectric effect can help us to determine some fundamental properties of superconductivity such as pairing properties and Andreev bound states. In this paper the authors study the work done by superconducting heat engine and the efficiency of the engine for bath temperatures ranging from 25 mK up to 800 mK. I'm wondering what specific applications there are for studying the work done by a heat engine and its efficiency at these extreme low temperatures. Because this journal has a broader readers, the authors should explain the significance of this work in detail.

We thank the referee for the positive comments on the manuscript. The results presented in this work represent a proof of principle device demonstrating and quantifying, for the first time, the heat to energy conversion in a superconducting tunnel junction. The extremely low operation temperature of the device directs its application area to the very specific environments of the dilution refrigerators. In this extreme environment we may already envision two possible application schemes: one for the local harvesting of the heat wasted by complex cold circuits, and another for high-sensitive bolometric radiation detectors. In both cases, the electrical energy generated could be re-used locally to power the system (the circuit or the readout circuit, respectively) without a detrimental thermal link to room temperature power sources.

Interesting applications may come by slightly increasing the operation temperature of the device above 3 K with the proper material selection as proposed in the new conclusions. In this temperature range the heat engine may be applied for energy harvesting in the deep space as briefly mentioned in the introduction.

We have modified a sentence in the conclusions to better explain the significance of our work for possible future applications in energy harvesting. See the Textdiff file lines 253-262.

2, The captions of figures 1 and 3 are too long and the author could have put some details into the text.

Captions length has been reduced by following the suggestion of the reviewer. (See the new captions in the Textdiff file)

3, Equation (1) is a bit too simple and too many parameters Δ, Γ, G, P and h are fitted through it, which will affect the accuracy of the results. The density of states of superconductors is theoretically calculated by self-consistency and varies with temperature, none of which is given in the paper. The authors can give the details in the supplementary material.

We note that Eq. (1) is the standard equation for the tunneling current, and its variants are used for example to explain majority of scanning tunneling spectroscopy experiments. It is correct that there are

some particular assumptions behind this expression, and that those assumptions cannot be fully checked from the fit. Rather, what we are looking for is consistency and checking with which parameters the fit to this expression is at best. The main assumptions are (i) tunneling limit, i.e., neglecting higher-order processes such as Andreev reflection, (ii) precise form of the superconducting density of states in the presence of both spin splitting and various relaxation processes, and (iii) rapid equilibration of energy inside both electrodes connected by the tunnel junction. These assumptions are rather standard and described in various reviews (see for example our Refs. 22 and 26).

Within the model, the parameters mentioned by the referee in fact are quite well specified and independent: Δ describes the average energy gap showing up directly in the subgap of the differential conductance, h describes its splitting (i.e., distance between the peaks) and P the asymmetry of the $G(V)$ peaks, Γ describes the size of the zero-bias conductance at low temperatures, and G the asymptotic differential conductance at high voltages. Small changes of these five fitting parameter will suddenly spoil the quality of the fit. In the manuscript we also mention the spin-flip parameter Γ_{sf} . It is somewhat harder to separate it from the phenomenological Dynes parameter Γ , but its role is to broaden the two spin-split peaks. In Ref. 22 (including some of the present authors) we also describe spin-orbit relaxation, but have neglected it here because Al as a light element has a weak spin-orbit coupling.

As for self-consistency: the fitted parameter is Δ_0 which is the value of Δ at zero temperature and $h = 0$. The value of $\Delta(T, h, \Gamma_{\text{sf}}, \Gamma)$ is indeed computed self-consistently from the theory of [22]. For the heat engine setup, the fitting considers experimentally obtained differential conductance in the temperature range 100–800 mK. All the fit parameters are assumed temperature independent, and have such values that the theoretical predictions for the dI/dV fit the experiments at all of the temperatures. We have now clarified these points in the main text (see the new method section B, line 302-304 of the Texdiff file).

We have added a new Extended Data Figure 4 that shows the theoretical dI/dV fits and Δ temperature dependence calculated by self-consistency.

4, Extended Data Figures 2 and 3 are given but are not illustrated or discussed in the paper or after Extended Data Figures 2 and 3.

Extended data figure 2 and 3 have been discussed in the main text at lines 90, 115, 138. We recognized that we wrongly recall them as "extended figures" and not "extended data figures" that probably has lead to the misunderstanding. We have correct this naming in the new version of the manuscript (See the Texdiff file).

In general, the authors have studied superconducting spintronic heat engine, which is very important and timely, but they need to give a more detailed and rational theoretical explanation of the results obtained.

We agree that the theoretical discussion of the heat engine aspect was in the previous version somewhat brief and therefore it was unclear where for example the low figure of merit originates from. Our Ref. 22 details an electrical circuit model for a thermoelectric device. This model shows that the maximum efficiency of a thermoelectric generator in the presence of an optimally chosen load resistor is $\eta = \eta_{\text{Carnot}}(\sqrt{1 + ZT} - 1)/(\sqrt{1 + ZT} + 1)$, where $\eta_{\text{Carnot}} = \Delta T/T_{\text{hot}}$ is the Carnot efficiency and ZT is the thermoelectric figure of merit of the device. For small ZT we hence get $\eta \approx \eta_{\text{Carnot}} ZT/4$. The thermoelectric figure of merit is obtained from

$$ZT = \frac{S^2 GT}{\tilde{G}^{\text{th}}},$$

where $S = \alpha/GT$ is the thermoelectric power, G is the conductance, α is the thermoelectric coefficient, T is the temperature and \tilde{G}^{th} is the total heat conductance across the thermoelectric contact at zero current. In the absence of electron-phonon coupling the total heat conductance is dominated by the tunnel junction $\tilde{G}^{\text{th}} \approx G_T^{\text{th}}$. For a low Dynes parameter and for $k_B T \ll h$, one can show (see Ref. 13) that the spin-polarized tunnel junction has a figure of merit that in the linear response regime is approximated by $ZT \approx P^2/(1 - P^2)$. With $P = 0.5$ and $\Delta T = 300 \text{ mK} \approx T_{\text{hot}}/2$, the corresponding ideal efficiency would be $\eta \approx 10^{-1}$. However, in our experiment non-idealities strongly decrease the engine efficiency, including in order of priority:

(i) the \tilde{G}^{th} dominated by electron-phonon coupling reducing the true ZT approximately by the ratio of the thermal conductance for the electron-phonon coupling in Co to that across the tunnel junction, i.e. $\gamma_{ph} \equiv G_{\text{e-ph}}^{\text{th}}/G_T^{\text{th}} \simeq 5At_{\text{Co}}\Sigma_{\text{Co}}T^4/(\pi^2k_B^2AG_{\square}T/(3e^2)) \approx (T_{\text{Co}}/8 \text{ mK})^3$. Here $\Sigma_{\text{Co}} \sim 2 \times 10^8 \text{ W}/(\text{m}^3\text{K}^5)$ is the e-ph heat conductivity estimated from dI/dV measurements, $G_{\square} = 20 \mu\text{S}/(250 \times 300 \mu\text{m}^2)$, and Co layer thickness $t_{\text{Co}} = 14 \text{ nm}$. We compare the electron-phonon heat conductance to the normal-state (Wiedemann-Franz) heat conductance of the tunnel barrier, because of the large subgap conductance (large Γ) of the samples, see Fig. 1d.

(ii) Additionally, reduction of ZT comes from the suppression of thermoelectricity by one order of magnitudes due to large Γ as estimated in the Extended data Figure 1.

(iii) Finally, the correction from superconductivity to γ_{ph} is at most one order of magnitude. All these factors gives at 400 mK a reduction of 5-6 orders of magnitude making $\eta \approx \frac{10^{-1}}{125000 \times 10 \times 10} \approx 10^{-8}$ in fair agreement with our findings.

The decrease of η as the temperature increases is mainly associated with the increase of the electron-phonon heat conductance.

In Fig. 3f, the saturation of η at lower temperatures is due to the Joule heating of Co, which keeps T_{Co} above $\sim 400 \text{ mK}$.

The main way to increase the efficiency would thus be reducing the (spurious) relative contribution of the electron-phonon coupling to the heat conductance. The primary method would be to heat the superconductor instead of the normal metal. This is not possible with our current setup and heating method, which involves relatively large-area tunnel junctions: a DC current through the superconductor would be converted into a supercurrent and therefore would not heat the electrons. We can note that this problem is not present in the absorption of high-frequency rf radiation. Heating of the superconductor with local current injection is in principle possible, but requires a different device structure.

Moreover, the tunnel junctions in our samples are rather opaque, and similar characteristics could be obtained also with more transparent junctions. In the electron-phonon dominated regime the efficiency is proportional to the junction conductance, so increasing it by some orders of magnitude while preserving its other characteristics would increase the efficiency by the same factor.

In addition the total heat conductance from the electrons via the phonons to the bath could be reduced by suspending the sample and thereby reducing the heat conductance of the phonons to the substrate.

In short, the relevant theoretical model is the combination of the thermoelectric effect described in Ref. 13, effect of the load described in Ref. 22, and the extra effect of electron-phonon coupling. The main factor limiting the efficiency (compared to Carnot) is thus the electron-phonon coupling and the high junction resistance, with some additional effects coming from slightly non-optimized load resistor and the finite value of the spin polarization of the junction.

We have now included the main ideas of this discussion in Sec. IV of the main text along with the end of the Methods section. (See lines 198-211 and 332-359 of the Texdiff file)

Referee 2

The current manuscript has studied on the superconducting spintronic heat engine consisting of a ferromagnetic insulator/superconductor/insulator/ferromagnet tunnel junction (EuS/Al/AlO_x/Co), which shows that the sign of thermoelectric voltage can be reversed depending on the parallel or anti-parallel states between ferromagnetic EuS and Co. Furthermore, the efficiency of the engine is evaluated for different magnetic configurations and thermoelectric memory cell has been discussed based on the device magnetic hysteresis. The theoretical model has been proposed to predict the engine efficiency with different parameters. Before providing my general opinion on the work, I wish to raise a few concerns:

1) The author claimed that a strong thermovoltage can be obtained at sub-Kelvin temperatures and the resulting Seebeck coefficient can reach up to the order of few hundred of $\mu\text{V}/\text{K}$ at different temperature and magnetic configurations. This should be the most important part for this work. However, such magnetic-order-dependent thermoelectric behavior have been discussed in the previous publications, such as Phys. Rev. Lett. 130, 237001 (2023) and Phys. Rev. Lett. 116, 097001 (2016).

The main novelty of our work, as highlighted by the title, is the thermoelectric heat engine implemented in a superconducting spintronic device. Differing from thermovoltage and thermocurrent experiments, already investigated in the seminal works of Beckmann (Phys. Rev. Lett. 116, 097001 (2016) as cited in refs [14,15]) up to the very last work of Ruano (Phys. Rev. Lett. 130, 237001 (2023), now refs [30,31]), here we measure, for the first time, the thermoelectric heat engine. This is an essential measurement that has never been directly performed despite to its importance to quantify the heat to energy conversion. Notably, such a measurement requires the presence of a load resistor that necessarily perturb the circuital scheme and may strongly affect the visibility of the thermoelectric effect conventionally measured in simpler open or closed circuital configurations for the thermovoltage and thermocurrent, respectively. Moreover, the control over a load resistor allows to quantify essential parameters of superconducting thermoelectric generators including their internal impedance and efficiency. The latter was surprisingly low despite to the large Seebeck coefficient observed also in similar devices and allowed to discover the detrimental impact of electron-hole heat losses present at sub-Kelvin temperatures.

These results are an important step forward in the understanding of superconducting heat engine, not directly accessible from indirect measurements of thermovoltages and thermocurrents of previous works.

Supplementary results, as recognized by the reviewer, are the spin-valve control over the the Seebeck coefficient and the implementation of a thermoelectric memory cell.

We thank the reviewer for suggesting the last work on thermoelectricity in superconducting spin valves (Phys. Rev. Lett. 130, 237001 (2023)) that having been published shortly before the submission of our article was missing in the list of references. We properly integrated in the new version of the manuscript by updating the related sentences, see refs [30,31] of the TexDiff file for the changes.

2) The authors have evaluated the engine efficiency for the so called the superconducting spintronic heat engine system, which is about 10^{-8} . The small energy conversion efficiency, in contrast to the conventional heat engine, would limit the further applications for the energy harvesting. Also, the discussion should be added to improve the working temperature by substituting superconductor Al with other high-temperature superconducting materials.

We agree with the referee, one of the important and surprising result of our research was the low efficiency of the heat engine respect to the original theoretical expectations. We identified such limitation mainly

in the strong electron-phonon dissipation (see also reply to referee 1) and proposed solutions in the conclusions.

We also agree on the quantification of the engine efficiency for higher-Tc superconductors like V or Nb that may be useful especially for future applications. Typically, high-temperature superconductors have larger spin orbit interactions additionally degrading the efficiency of the engine and so that special attentions need to be taken when designing the engine. Moreover, at higher temperature the electron-phonon shunting heat channel is stronger. An optimal heat engine operating at 3 K have been evaluated. Thin Nb or V films replaces the Al of our device.

By first neglecting the spin-orbit relaxation and assuming $\Gamma \ll \exp(-\Delta/k_B T)$, $k_B T \ll h \ll \Delta$ the figure of merit ZT can be estimated from Eq. (151) of ref [22]. With $t_{Co} = 14 \text{ nm}$, $P = 0.5$, $\Sigma_{Co} \sim 2 \times 10^8 \text{ W}/(\text{m}^3 \text{K}^5)$ as for our experiment, $G_{\square} \sim 2 \times 10^5 \text{ S}/(\text{m}^2)$ thousand times larger than in the experiment, $h = 0.3\Delta$ and $\Delta_{Nb} = 1.3 \text{ meV}$ we estimate $ZT \approx 0.004$ and hence $\eta \approx 0.001\eta_{\text{Carnot}}$. Introducing a spin-orbit scattering time $\tau_{so} = 0.2 \text{ ps}$ typical for Nb (Phys.Rev.Lett.112(3)(2014) 036602), there is a further factor 10 reduction due to spin-orbit scattering as can be estimated from Fig. 30 of ref [22] with $\hbar/(\tau_{so}k_B T_{c0}) \approx 4$. Note that this scheme would require the presence of quite large exchange splitting in Nb, equivalent to fields of order of several T. Indication of such large exchange fields have been obtained in Nb/GdN and Nb/EuS bilayers (Phys. Rev. B 108(14), L140502 (2023); Phys. Rev. B 92(18), 180510 (2015); Nat Commun 14(1), 1630 (2023))

Heating of the superconductor with local current injection is in principle also possible, given a different device structure than used in this work. In that case the role of electron-phonon scattering is strongly reduced, and the efficiency is limited by the subgap density of states, only. In this case is possible to reach $ZT \approx 0.1P^2/(1 - P^2) \approx 0.04$ and efficiency $\eta \sim 0.01\eta_{\text{Carnot}}$, see Fig. 30 in ref [22].

we have included this useful discussion in the conclusion of the manuscript together with the estimation for high-Tc superconductors in the methos section (see lines 253-262 and 333-359 of the Textdiff file)

Other minor comments:

- *The labels in the extended data Fig. 2 should be distinguished.*
- *The first letter of journal name in the references should be capital.*
- *The units and the number should be separated.*

- Labels and format of extended data figure 2 and 3 adjusted to fulfill the journal requirements
- Typos in the capital letter of the journal corrected
- Missing separations introduced.

Overall, the results are important but some of them are not completely new in the paper, although they are valid. Therefore, I cannot recommend for the publication in Nature Communications.

We thank the reviewer for recognizing the validity of our results. As explained above, the main result of our work is the demonstration and quantification of the heat engine, that, to the best of our knowledge, has never been realized. The measurements of a thermovoltage or thermocurrent already observed from the 2016 and based on open or closed circuits are not sufficient for such a demonstration.

Referee 3

The manuscript proposes a fabrication and also theoretical investigation of the superconducting spintronic heat engine consisting of a ferromagnetic insulator/superconductor/insulator ferromagnet tunnel junction (EuS/Al/AlOx/Co). Also, the implementation of the thermoelectric memory cell is reported due to the reverse of the sign of the generated thermoelectric voltage according to the parallel or anti-parallel orientation of the two ferromagnetic layers. The work is interesting specially for thermoelectric-based applications in cryogenic electronics where at the extremely low temperatures, the semiconductor materials are not anymore suitable for energy harvesting.

Although the work sounds, but it lacks perfection considering the following concerns:

- The lack of commas “,” is felt throughout the whole manuscript, leading to the misdirect concepts. For example: P4, line 61: A sizable work was extracted by the junction therefore demonstrating a superconducting spintronic heat engine. This sentence without any comma is misleading and hard to follow

We thank the reviewer for the positive comments on our results and for the suggestion to improve the readability of the text. Changes are visible and underlined in the new version of the text.(see the textdiff file for a direct comparison)

- Very long sentences:

For example: P4, lines 82-87: By fitting the experimental data with the spin selective tunneling model [22] (see Methods section VII B for model details and continuous lines in Fig. 1d for the fit) it is possible to extract the spin polarization of the tunnel barrier $P \simeq 0.5$, the exchange interaction induced in the Al layer $h \simeq 50\mu\text{eV}$, the Al superconducting gap $\Delta \simeq 195\mu\text{eV}$, and the inelastic and spin-flip scattering rates $\hbar\Gamma \simeq 32\mu\text{eV}$ and $\hbar\Gamma_{sf} \simeq 29\mu\text{eV}$, for $B = -35\text{mT}$, respectively.

Long sentences have been split to improve readability. (see the main changes in the texDiff file)

- Verification/validation Cases:

*For example: P.5, lines 93-95: Values obtained are compatible with previous TMR measured in similar structures [23], showing a maximum at voltage biases compatible with the superconducting gap ($eV_{\text{MAX}} \simeq \pm\Delta$). It is important that the author name the specific case of the verification clearly and mention the approach. It is better to show the obtained result and the case of the verification/validation in the same plot, not just citing the work. In the present form one should go and check the reference [23] and find out that only the behavior is the same while the authors claim the same structure is investigated. [23] B. Li, G.-X. Miao, and J. S. Moodera, Observation of tunnel magnetoresistance in a superconducting junction with zeeman-split energy bands, *Physical Review B* 88, 161105 (2013).*

Or the case of validation also from the present manuscript: P 5,6, lines 119-121: Hysteresis in the magnetic field is visible, with a maximum signal at $|B| \simeq 20\text{mT}$ vanishing above 120 mT due to the quenching of superconductivity, as observed also in the tunneling spectroscopy measurements reported in fig. 1c. In fact, the manuscript lacks the strong validation/verification part. To make the manuscript clearer and more reliable, I suggest the author

write a section entitled “verification/validation” to mention the cases. The results and the validation/verification cases are mixed and this makes the manuscript difficult to be followed.

The data of fig.1 mentioned by the referee are electrical pre-characterization of the device introduced to evaluate the physical parameters used in the engine modeling. The recall to the device presented in ref [23] is just pedagogical without any claim of a quantitative comparison. In fact, while the two devices are similar in the material layers, they differ in the thicknesses making the one-to-one comparison not significant and not necessary for the scope of the work.

To better address the scope of fig.1 as pre-characterization we specified it in the title of the figure. (see textdiff file)

Regarding the validation/verification case between fig.2c and fig.1c we agree with the referee, it may be very difficult to compare a linear-plot of thermovoltage with a color-plot of differential conductance. For that reason we extended the explanation of figure 2c and we use the magneto-conductance of fig.1f stress the analogies between the two hysteretic features. (see textdiff file, line 129-130)

- Some sentences are needed to be more explained The manuscript presents lots of information but somewhere it lacks the clear justifications. For instance,
1. The behavior of the efficiency appeared in Fig. 3f is not well justified.

The behaviour of the efficiency and its relation with the intrinsic parameters of the device is now better explained in the main text, see also the reply to Referee 1 above for a detailed discussion and the Textdiff file at lines 198-211 and the new method section at lines 333-359.

2. I do not see a clear explanation about the claim that “heat losses through the electron-phonon coupling as the main factor limiting the engine efficiency”.

In the new text we provide estimates of the reduction in the efficiency from various source, see also reply to Referee 1 for more details. In short we quantified the reduction of the efficiency in the:

- (i) electron-phonon leak channels for $\approx 10^{-5}$
- (ii) non-ideal Γ for $\approx 10^{-1}$
- (iii) correction from superconductivity for $\approx 10^{-1}$

For the main changes see the Textdiff file at lines 198-211 and the new method section at lines 333-359.

3. More explanation of the claim “by scaling their heat engine to the large areas, they may find relevant applications for energy harvesting in the deep space”. How does this work?

The proof of principle device we present in our work is based on an additive property of the surface of the tunnel junction. Therefore, despite to the small power produced in our experiment (\leq fW), the scheme could be in principle scaled to large area to produce the desired power for the specific application. In the particular application proposed for energy harvesting in the (cold) deep space it would be necessary to increase the operation temperature of the engine above 3K. We modified the main text proposing in the conclusion alternative materials that could fulfill this requirement (see lines 253-262 of the textdiff file) and realistic designs to improve the engine efficiency up to $0.01\eta_{\text{Carnot}}$ (see lines 343-359 of the textdiff file).

In brief, I recommend the publication of the manuscript if the mentioned comments are fulfilled.

We thank the reviewer and hope that our reply fulfills all their requirements.

Changes in the manuscript

In response to the referees, we have made various changes to the manuscript text. Besides additions requested by the referees, we have improved the grammar and style of the text in various places. Due to the difficulties in listing all the changes the differences between the old and new versions of the manuscript are detailed in the associated file Textdiff.pdf. Figure 3, Extended data figure 2 and 3 updated, and Extended data figure 4 included.

REVIEWERS' COMMENTS

Reviewer #1 (Remarks to the Author):

The revised manuscript takes full account of all the points raised in the previous reports. I recommend the manuscript for publication in Nature Communications in its present form.

Reviewer #2 (Remarks to the Author):

All my concerns are well addressed. Thus, I recommend it for publication in Nature Communications.

Reviewer #3 (Remarks to the Author):

The manuscript in the present form can be accepted for the publication.